# Diet-Induced Models of Non-Alcoholic Fatty Liver Disease: Food for Thought on Sugar, Fat, and Cholesterol

**DOI:** 10.3390/cells10071805

**Published:** 2021-07-16

**Authors:** James M. Eng, Jennifer L. Estall

**Affiliations:** 1Institut de Recherches Cliniques de Montréal (IRCM), Montreal, QC H2W 1R7, Canada; james.eng@ircm.qc.ca; 2Faculty of Medicine, University of Montreal, Montreal, QC H3T 1J4, Canada

**Keywords:** NAFLD, NASH, liver, dietary model, high-fat diet, cholesterol, liquid sugar, solid sugar

## Abstract

Non-alcoholic fatty liver disease (NAFLD) affects approximately 1 in 4 people worldwide and is a major burden to health care systems. A major concern in NAFLD research is lack of confidence in pre-clinical animal models, raising questions regarding translation to humans. Recently, there has been renewed interest in creating dietary models of NAFLD with higher similarity to human diets in hopes to better recapitulate disease pathology. This review summarizes recent research comparing individual roles of major dietary components to NAFLD and addresses common misconceptions surrounding frequently used diet-based NAFLD models. We discuss the effects of glucose, fructose, and sucrose on the liver, and how solid vs. liquid sugar differ in promoting disease. We consider studies on dosages of fat and cholesterol needed to promote NAFLD versus NASH, and discuss important considerations when choosing control diets, mouse strains, and diet duration. Lastly, we provide our recommendations on amount and type of sugar, fat, and cholesterol to include when modelling diet-induced NAFLD/NASH in mice.

## 1. Introduction

Non-alcoholic fatty liver disease (NAFLD) is a common condition with rising prevalence that has, in recent years, become a global public health concern [1,2]. The high prevalence of NAFLD is linked to overnutrition and consumption of highly processed foods. The disease is characterized by aberrant accumulation of lipids in hepatocytes and is highly correlated with obesity, insulin resistance, and type 2 diabetes [2]. Consequently, NAFLD is widely considered to be the hepatic manifestation of metabolic disease and encompasses a spectrum of liver pathology [3,4,5]. If left untreated, NAFLD progresses to involve hepatic inflammation and early fibrosis, which mark the transition to non-alcoholic steatohepatitis (NASH) [4,5]. End stages of NAFLD include inflammation, fibrosis, and/or cirrhosis of the liver and increased risk for hepatocellular carcinoma (HCC) [4].

One major hurdle to study NAFLD in a laboratory is the lack of appropriate animal models in which to investigate the disease. Until recently, there has been widespread belief that replicating the pathological features of human NASH in animal models is difficult or impossible without use of hepatotoxins, nutrient-depleted diets (e.g., methionine-choline-deficient diets) or macronutrient composition sometimes far from what humans normally ingest. Obesogenic diets similar to modern human consumption can induce pathology, yet often require over 16 weeks to initiate the transition from NAFLD to NASH in rodents. These “milder diets” increase duration and cost of in vivo studies, providing incentive to develop dietary models that reliably and quickly model NAFLD and NASH, while also maintaining relevance to human pathology.

Newer dietary protocols often contain high levels of fructose or sucrose, high-saturated-fat content, and cholesterol; however, it is also still common to study NASH in models using micronutrient-deficient diets or hepatotoxins to expedite and exaggerate liver pathology. Unfortunately, the latter may result in poor translation of results to human pathophysiology, decreasing confidence in pre-clinical models [6]. Considerable differences between murine NAFLD/NASH models also make it difficult to compare results between studies and over time. In this review, we present new research findings on the impact of rodent diet on NAFLD pathophysiology, focusing on recently published data where there is renewed interest to refine dietary models to be more physiologically relevant to human NAFLD and maximize translatability. We focus on comparing individual roles and contributions of dietary components and discuss common misconceptions surrounding frequently used diet-based NAFLD models.

## 2. NAFLD, NASH and MAFLD

Although widely used, the term NAFLD is criticized by some over its vague definition and misleading nomenclature. NAFLD is traditionally classified by the presence of fatty liver (>5% of hepatocytes with steatosis) in the absence of other hepatic conditions, such as viral hepatitis or excessive alcohol consumption [7]. This implies that NAFLD is only present when other pathologies are ruled out, which is misleading, since NAFLD can coexist with other liver diseases [8]. The alternative term, metabolic-associated fatty liver disease (MAFLD), was recently suggested, reflecting metabolic dysfunction as the main driver of this variety of liver disease [7].

With this in mind, four main characteristics associated with metabolic dysfunction are the goal of animal models of MAFLD, including insulin resistance, hepatic steatosis, fibrosis, and inflammation. Individually, these can be achieved in many ways, including genetic or chemical disruption of metabolic processes (e.g., CCL_4_ treatment), but it is generally accepted that MAFLD in humans is closely linked to systemic metabolic dysfunction, poor nutrition and a sedentary lifestyle. Therefore, there is increasing resistance to using knock-out mice, models containing hepatotoxins, or diets that cause liver damage in the absence of metabolic dysfunction in order to maximize the translatability of animal model to clinical practice.

There is now abundant research surrounding the role of macronutrient type, form and source in metabolic liver disease pathobiology. We compare the roles of carbohydrate, dietary fat and cholesterol in modulating NAFLD severity, and highlight the influence of mouse strain. We also present our recommendations and outline important experimental design considerations when one’s goal is to carry out pre-clinical modeling of MAFLD.

## 3. Fructose, Glucose, and Sucrose: Is There Really a Difference?

It is well accepted that sugars, while an essential source of energy, can have detrimental effects on liver health. Fructose is a major dietary monosaccharide commonly found in food as part of the disaccharide sucrose (table sugar, composed of one glucose molecule and one fructose) or as a component of high-fructose corn syrup (mixture of fructose with sucrose or glucose) [9,10]. Fructose has been extensively studied in the context of liver disease, obesity, and diabetes over many decades and is known to promote hepatic de novo lipogenesis, lipid accumulation, and insulin resistance [9,11,12]. In contrast to glucose, fructose is almost entirely cleared from circulation by the liver via the GLUT5 transporter, where it can bypass glycolysis, the rate-limiting step in acetyl-CoA production. Thus, a large quantity of acetyl-CoA is produced quickly following fructose uptake [13,14]. Some acetyl-CoA is used for ATP production within the citric acid cycle; however, this cycle quickly becomes overwhelmed and the remaining acetyl-CoA is shuttled into de novo lipogenesis pathways [13].

While most animal research related to sugar in NAFLD revolves around fructose, it is not the only sugar added to processed food in high quantities. Fructose, glucose, and sucrose are the most consumed sugars; but it is only starting to be recognized how these sugars compare with regard to their hepatotoxic properties. Comparing 30% (*w*/*v*) fructose water to 30% (*w*/*v*) glucose water, Softic et al. found that each monosaccharide has different effects on weight gain, metabolic phenotype, and gene expression. Fructose is more effective than glucose at inducing obesity, glucose intolerance, and insulin resistance when combined with a high-fat diet (60% of total kcal from fat) in mice [10]. Excess fructose increases *Srebp1c* mRNA, while excess glucose mainly increases *Chrebp* mRNA levels [10], suggesting that fructose favors lipogenesis while glucose is glycolytic in addition to lipogenic [15]. Similarly, in Wistar rats, a high-fat/high-fructose diet stimulates greater increases in plasma insulin and glucose, compared to high fat-only or high-fat/high-glucose diets [16].

Even though fructose is a component of sucrose, fructose alone seems to have differential effects on protein metabolism compared to sucrose. For example, fructose and sucrose differ in their ability to modulate specific amino acids in serum [17]. The authors hypothesize that this difference is due to discrepancies in the activity of various transaminase enzymes, with increased conversion of fructose to alanine, serine, glycine, and cystine compared to sucrose [17]. A study by Sadowsk and Bruszdowska comparing sucrose to 55% high-fructose corn syrup (HFCS-55, 55% fructose:45% glucose) in Wistar rats over 6 weeks shows no difference between the two sugar preparations with regard to weight gain or energy intake, but HFCS-55 also increases plasma glucose and triglycerides [18]. Mock et al. found that rats drinking high-fructose corn syrup have the highest level of hepatic de novo lipogenesis and steatosis, and the greatest reduction in β-oxidation, compared to fructose or sucrose sweetened water [19]. These results suggest that the unique combination of free glucose and fructose in HFCS-55 may be more effective at stimulating de novo lipogenesis and increasing liver steatosis than when the monosaccharides are bound together as sucrose. These studies also suggest that the combination of fructose and glucose may be more hepatotoxic than fructose alone, but this may be because rats consume significantly more high-fructose corn syrup (and total calories) compared to fructose alone [19,20,21].

Similar results are seen when different sugars are given in solid form. Sánchez-Lozada et al. compared solid sucrose (60% kcal) versus a 50:50 solid glucose:fructose mix (30% kcal glucose, 30% kcal fructose) and found that the mix of glucose + fructose produces higher hepatic triglyceride levels compared to sucrose after 16 weeks [12]. However, the sucrose-fed group have higher hepatic levels of the cytokines monocyte chemoattractant protein 1 (MCP1) and tumor necrosis factor-alpha (TNFα) [12], suggesting worsened inflammation. Interesting, Andres-Hernando et al. show that a mix of glucose and fructose monosaccharides accelerates absorption of fructose into the liver following oral administration [20], suggesting a mechanism by which glucose may exacerbate the hepatotoxic effects of fructose. These results highlight differences in physiological response to dietary sugars and indicate that not all high-sugar diets are equivalent. They also imply that a mixture of fructose and glucose may be more effective at inducing hepatic steatosis compared to sucrose or fructose alone, whereas sucrose seems to more strongly promote hepatic inflammation.

## 4. Solid vs. Liquid Sugar: Which Is Better for Promoting NAFLD/NASH?

With regard to how sugar is added to a diet (e.g., in drinking water or solid food), there is lack of consensus on which medium is best or whether it actually matters. Humans consume a significant proportion of their sugar in liquid form, with an average of 6.5% of total daily calories coming from sweetened beverages for adults in the United States [22]. Increased consumption of sweetened beverages is associated with increased risk of NAFLD [23,24], indicating that liquid sugars merit careful consideration in the context of NAFLD. A meta-analysis performed by Toop and Gentili reviewed 26 studies, varying in duration between 3 and 38 weeks, investigating the effects of dietary fructose in rodents. They found that concentrations of between 10% and 21% (*w*/*v*) liquid fructose are associated with increases in weight gain, blood glucose, insulin levels, and plasma triglyceride levels compared to control animals [25]. They also conclude that liquid fructose is sufficient to induce early indicators of metabolic syndrome and is a more relevant dietary regiment than excessive solid fructose (>60% *w*/*w* content) [25]. Other studies show that 15% and 30% fructose water increase plasma cholesterol and worsens oral glucose tolerance in mice, with 30% fructose also increasing body mass, fasting blood glucose, and plasma triglyceride levels after 9 weeks [26]. Liquid fructose may permit the use of lower overall concentrations and avoids use of unrealistically high amounts of simple sugars in solid food that do not reflect human diets. However, the question remains as to whether it is simply a dosage argument, or whether the pathology of NAFLD/NASH differs due to the medium of sugar delivery.

A study comparing mice fed a high-sucrose diet with access to liquid sucrose (30% kcal solid, 50% *w*/*v* liquid) versus a matched solid diet high in solid sucrose (73% kcal) demonstrates that feeding a combination of liquid and solid sucrose increases sugar ingestion as well as total caloric intake, correlating with increased gain in body fat over 8 weeks [27]. Importantly, while their intention was likely to compare sugar 100% in solid form versus 50% in solid + 50% liquid while matching energy intake, the 100% solid sugar group had lower total sugar intake. Overall, consumption of a proportion of sucrose in liquid form results in greater adiposity, increased plasma glucose, fasting insulin, and hepatic lipids compared to the group consuming 100% solid sugar [18,27]. Similarly, Mastrocola et al. found that with matched fructose intake, liquid fructose (60% *w*/*v*) more potently promotes hepatic steatosis and fibrosis over 12-weeks, whereas solid fructose (60% kcal) promotes a stronger inflammatory response in liver (Figure 1) [28]. There is also evidence that suggests liquid and solid sugars are metabolized differently and evoke different physiological responses [28,29]; however, more research is needed to better understand these differences. In the interest of maintaining relevance to human pathology and consumption patterns, a combination of solid and liquid sugars might be preferred.

## 5. Is High Sugar Enough?

The average added sugar consumption level in the United States between 2007 and 2008 is reported to be 78 grams per day, accounting for approximately 15% of total daily caloric intake [30]. Of this added sugar, it is estimated that approximately 40% is fructose and the remaining 60% is sucrose [31]. High sugar consumption is known to increase the risk of NAFLD [24]; however, it is still unknown whether high-sugar diets alone are sufficient to fully drive NAFLD pathology. In Wistar rats fed a low-fat diet (3% kcal fat), supplementing with 10% sucrose or 10% high-fructose corn syrup, the added sugar does not affect fasting blood glucose, insulin levels, weight gain, or adiposity in the short term (6 weeks) [32].

This suggests that feeding sugar alone or fat alone (see below) might not be sufficient to promote NAFLD, and that the combination of macronutrients is what drives pathology. Differences in metabolic adaptation and mitochondrial function are noted when comparing high fat only- and high-fat + high-sucrose-fed mice. A high fat-only diet promotes glycolysis, while addition of sucrose to the diet decreases glycolysis [33]. Addition of 30% sucrose water to a high-fat diet (30% fat) decreases oxidative phosphorylation in liver and reduces mitochondrial replication [34]. Like in chow-fed mice, sucrose increases de novo lipogenesis when added to a high-fat diet [33,35,36]. Addition of solid sucrose (34% *w*/*w*) to a high-fat diet (60% kcal fat) may also be critical to transition to the NASH phenotype, as a high-fat diet without sucrose promotes steatosis but fails to convert to NASH [33]. In contrast, ingestion of 30% sucrose in water can induce liver damage and increase hepatic steatosis [35,36], and even higher concentrations (40% and 50%) can promote hepatic inflammation and fibrosis [35]. However, maybe this is just too much sugar? As low as 4.2% sugar water (42 g total sugars/L: 23 g/L fructose and 19 g/L sucrose) exacerbates glucose intolerance and insulin resistance induced by a high-fat diet (40% kcal fat) [21,37], indicating that even modest additions of liquid sugar can have big effects. Thus, the proportion of simple sugar required to promote NAFLD/NASH is still unclear and seems to depend heavily on the content of other dietary components.

## 6. The Great Fat Debate

It is well established that diets high in fat cause obesity and hepatic steatosis [38]. However, there is little consensus on how much fat is appropriate in rodent diets to promote NAFLD/NASH. Hu et al. show that, in mice, with increasing dietary fat, total energy intake increases proportionally until the diet is 50% fat, where it then reaches a plateau [39]. Adiposity is similar, with proportional increases up until diet is 60% fat [39,40]. They also show that dietary fat, not carbohydrate or protein, is the main driver of increased energy consumption and increased adiposity, demonstrating the importance of high-fat content in diets intended to generate obesity [39].

For NAFLD, a range of fat content from 32% to 60% fat is commonly used in high-fat diets to promote liver disease [41]. There is also significant heterogeneity with regard to the type of fat used, with lard (a largely saturated animal fat) being a common source often used in combination with soybean oil, a mix of saturated and unsaturated lipids [41]. Addition of soybean oil appears to enhance cholesterol-mediated damage to mitochondrial function, increasing oxidative stress, inflammation, and fibrosis in liver [42,43]. Given the wide variety of dietary fat concentrations and sources used in rodent diets, and the general lack of studies controlling or comparing them, it is difficult to recommend an appropriate or optimal dosage necessary for fatty liver disease.

Diets of obese individuals with NAFLD are typically around 36% fat, 46% carbohydrate and 18% protein [24]. The rising popularity of ketogenic diets (30% protein, 10% carbohydrate, 60% fat,) for weight loss in humans has recently brought attention to the importance of macronutrient ratios. Diets very high in fat (55% or higher) can promote ketogenesis depending on other macronutrient ratios [44], which is not typically associated with NAFLD/NASH and may be counterproductive to obesity [45,46]. Despite this, diets that are 60% fat (20% protein, 20% carbohydrate) are commonly used to promote obesity in rodents, but might not be optimal. Recent evidence suggests that a better balance of high fat and high carbohydrate (with fructose in some form) is preferable to a high fat-only diet. With quantities of fat and sugar set within a reasonable range, researchers turn their focus to other micronutrient factors that seem to have large impacts on NAFLD and NASH.

## 7. Cholesterol: A Question of Quantity?

It is already established that dietary cholesterol has an important role in the progression of NAFLD to NASH by inducing inflammation and fibrosis [47,48]. Only a small increase in cholesterol (0.06%) is needed to detect an increase in plasma cholesterol in mice [49], demonstrating its potent effect, even in small doses. However, much more seems to be needed to see pathogenic effects in mouse liver tissue. Ioannou and colleagues tested varying levels of cholesterol in combination with a higher-fat diet (15% fat) in mice over 6 months. They found that dietary cholesterol has a dose-dependent effect on murine liver cholesterol content and observe signs of steatohepatitis and fibrosis at concentrations above 0.5% [50]. They note that crown-like structures are present in the liver at 0.5% dietary cholesterol and above, and mRNA levels of proinflammatory genes are increased, reaching maximal induction at 0.75% cholesterol [50]. These results suggest that a minimum 0.5% dietary cholesterol may be necessary to promote NASH in rodents, and concentrations greater than 0.75% may not be warranted. However, as we have learned, pathogenic nutrients tend to have synergistic effects and this study was performed using a diet only moderately high in fat. Inclusion of sugar could further promote the profibrotic and proinflammatory effects of cholesterol.

Inclusion of a modest amount of cholesterol (0.2%) in the diet, in combination with high fat and fructose, worsens hepatic steatosis and inflammation, in addition to exacerbating blood glucose and insulin resistance after 14 weeks [51]. 0.2% added cholesterol also increases hepatic tumor rate compared to high-fat diet alone after 26 weeks feeding [52]. Addition of 0.75% cholesterol to a high-fat diet further increases triglyceride and cholesterol levels in serum and liver tissue [53], and increases hepatocyte ballooning, hepatic TNFα levels, and NAFLD activity score (NAS), indicating greater hepatic damage [54]. Impacts of 0.75% dietary cholesterol on hepatic inflammation and fibrosis are markedly increased when combined with poly-unsaturated fatty acids (PUFA), like ω6-PUFA from soybean oil [42,53], whereas this amount of added cholesterol also causes more severe steatosis, weight gain, and insulin resistance [43]. In comparison, we show that mice fed a high-fat/high-fructose diet supplemented with 2% cholesterol (40% kcal fat) have increased inflammation and fibrosis in liver, while obesity, insulin resistance and glucose intolerance are less pronounced compared to mice fed a similar diet without cholesterol [55]. Consistently, mice fed obesogenic diets with higher cholesterol levels, such as 1.25% and 2%, seem to exhibit similar impairments in weight gain [55,56,57]. Thus, too much cholesterol may counteract the goal of modeling liver damage within a setting of obesity and insulin resistance.

Studies in guinea pigs reveal that 0.08% cholesterol content is equivalent to human intake of 600 mg/day [58], while the average North American consumes closer to half this amount [59]. Many “Western”-type rodent diets include up to 2% cholesterol, arguably much greater than that found in human diets. This translates to approximately 15,000 mg/day in humans, 50-fold over recommended daily intake [58] and raises concern that this level of cholesterol in rodent chow is simply hepatotoxic and not representative of human diets linked to NAFLD/NASH. It is interesting to point out that there are strong associations between high-cholesterol intake and cancer in humans [60]. Using the HCVcpTG mouse, which spontaneously develops liver tumors due to expression of viral proteins, Wang et al. show that a diet containing 1.5% cholesterol increases incidence of liver tumors from 41% to 100% after 15 months [61]. Rodent studies also show that dietary cholesterol potentiates liver cancer when combined with obesity [52]. Interestingly, DNA mutations found in liver tumors of high-fat, high-cholesterol-fed mice are more numerous and more oncogenic than those in mice fed high-fat diet without cholesterol, and many of the same mutations found in cholesterol-fed mice are also found in human HCC [52]. Since NAFLD and NASH are both clearly associated with increased liver cancer risk [62,63], one may even argue that added cholesterol is an essential dietary component to model human disease.

If cholesterol is essential, but we want to avoid confounding toxicity, what is the appropriate amount to use in rodent chow to balance all aspects of the metabolic syndrome? Available data suggest that to model obesity-driven NAFLD/NASH and related HCC, it is advisable to not exceed 1% cholesterol content, due to the negative effects of higher levels on weight gain. A cholesterol content of 0.5% seems sufficient to facilitate development of steatohepatitis and fibrosis within a reasonable time frame, and anywhere between 0.2% and 1% cholesterol promotes development of spontaneous hepatic tumors. While an amount between 0.2 and 1% still appears much higher than levels normally ingested by humans, data in mice also suggest that differences in rodent versus human cholesterol absorption and metabolism may underlie need for higher amounts in rodent chow to model human disease [64,65,66].

## 8. Is Trans Fat the New Cholesterol?

The role for dietary cholesterol in cardiovascular disease has recently been downplayed [58,67], and trans fat became a new enemy. Trans fats are a form of unsaturated fatty acids where the carbon chain double bond is in the *trans* configuration. Trans fats are not naturally abundant but commonly produced by hydrogenation of vegetable oils [68]. Inclusion of trans fats in rodent high-fat diets increase hepatic insulin resistance, serum alanine aminotransferase (ALT) levels, hepatic steatosis, and expression of lipogenic genes when compared to a similar non-trans fat containing diet, demonstrating that trans fat can have more potent hepatotoxic effects [69,70,71]. Comparing a Western diet containing trans fat (40% kcal fat, 30% kcal Primex shortening) to a standard high-fat diet devoid of trans fats (60% kcal largely lard fat), Komastsu et al. noted that both diets have comparable levels of steatosis, but trans fats reduces storage of lipids in adipose tissue [57]. However, a recent study challenges the notion that *trans* fats are more harmful than non-trans fat. Drescher et al. show that non-trans fats in a Western diet fed over 24 weeks leads to more pronounced weight gain, glucose intolerance, increased ALT and fibrosis compared to a matched Western diet containing trans-fat [72]. Despite conflicting data, there remains sufficient concern that trans fats are detrimental for metabolic health. This has caused a problem for researchers trying to model NAFLD and NASH in the lab.

The Food and Drug Administration (FDA) banned the addition of trans fats to human food products, which created a scarcity of the trans-fat source for rodent chow, and manufacturers and researchers were forced to seek out alternative sources of fat with equal pathogenicity. To this end, Boland and colleagues compared a Western diet containing 40% kcal fat (trans fat-free saturated fat) to the amylin (AMLN) diet (40% kcal fat, 22% trans-fat and 26% saturated fat by weight). By week 16, the non-trans-fat-fed mice have higher weight gain and worse glucose tolerance compared to mice fed trans fat, but were comparable in liver weight, steatosis, and fibrosis. At 28 weeks, both diets produced similar levels of fibrosis, steatosis, and inflammation indicating that that the non-trans fat containing diet is a potential alternative to the AMLN diet to model NASH [73]. Thus, at least in mice, trans fat might not be the enemy that we thought it was, turning our attention toward other pathogenic fat types (e.g. saturated fats) instead.

Taken together, recent data confirm that diet-induced models of NAFLD are complex and phenotypes are highly influenced by the type and quantity of sugar, fat, and cholesterol included in the diet. However, based on data reviewed so far, recommendations can be made to improve relevance and translatability of rodent models for the pre-clinical NAFLD/NASH research community (Figure 2).

## 9. Matched Control Diets: What Should I Be Using for “Normal” Chow?

One aspect of animal study design often overlooked is the choice of the control diet. The control serves as the baseline to which all results are compared, yet this diet is often an afterthought, influenced by convenience or budgetary concerns. Rodent control diets are often high in carbohydrate, but the form of carbohydrates is an important variable. Gonzalez-Blázquez et al. recently compared a standard animal facility chow diet (24% kcal protein, 58% kcal carbohydrate, 18% kcal fat) to a low-fat control diet (18% kcal protein, 71.8% kcal carbohydrate, 10.2% kcal fat) matched in macronutrient source to the supplier’s high-fat diet [49]. In addition to being higher in total carbohydrates, the matched control diet is largely composed of refined sugars (sucrose, dextrin, and maltodextrin) compared to the crushed grain, wheat and legume used in the standard animal facility chow diet [49]. The matched control diet also contains lower levels of PUFAs and slightly higher cholesterol content (under 0.1%) [49]. Interestingly, mice fed the matched diet have higher energy intake, plasma insulin, glucose, cholesterol, and triglycerides compared to the chow-fed mice after 6 weeks [49]. The authors attribute differences between the two “control diets” mainly to (1) higher carbohydrate content, (2) the larger proportion of refined sugars, and (3) presence of cholesterol in the matched control diet. Thus, while the matched control may be the best control to model the effects of specific macronutrients (i.e., fat), it may not actually represent a “healthy” diet for a mouse.

In contrast, work from our research group shows that a common grain-based chow diet (24% kcal protein, 58% kcal carbohydrates and 18% kcal fat) can surprisingly promote weight gain and hepatic steatosis similar to a high-fat/high-fructose diet (20% kcal protein, 35% kcal carbohydrates, 45% kcal fat, supplemented with 30% fructose water) when mice are fed over long periods of time (8 months) [55]. Because the standard chow diet is obesogenic on its own, the metabolic effects of the high-fat/high-fructose diets are significantly masked [55]. Accumulating literature provides evidence that many standard lab chow formulations are not “healthy” and can change ingredient sources batch-to-batch. This demonstrates that the choice of control diet can be just as important as the choice of pathogenic diet in experimental design. Careful consideration of experimental goals should inform this decision, choosing whether you desire a healthy control diet or you want to control for specific diet components.

## 10. Is Mouse Strain Important?

Heterogeneity across mouse strains is a constant struggle when comparing results across studies. Generally, C57BL/6 mice are used for studies in metabolic disease and NAFLD/NASH. Comparing male C57BL/6N, CD-1 and 129Sv mice, C57BL/6N mice gain more weight on a high-fat diet, but can also have a heterogenous response to diet-induced NAFLD [74]. While 129Sv mice do not become obese, they develop hepatic steatosis and inflammation, but show little hepatocyte ballooning. In contrast, CD-1 mice do not develop metabolic or hepatic markers of NAFLD [74]. While C57BL/6 mice are tremendously popular, recent data show that not all C57BL/6 mice are equivalent.

A study comparing C57BL/6J versus C57BL/6N mice identifies significant differences between these two closely related substrains. Due to drift, C57BL/6J and C57BL/6N mice have differences at the genomic level, including over 30 single nucleotide polymorphisms (SNPs) [75]. Among these genetic differences is a mutation in the *Nnt* gene in C57BL/6J mice, which impairs insulin secretion and glucose tolerance [76,77]. Kawashita and colleagues compared the response of C57BL/6J and C57BL/6N mice to CCl_4_, a toxin that causes significant liver damage. They note that C57BL/6J mice exhibit more substantial fibrosis and oxidative stress following CCl_4_ treatment compared to C57BL/6N mice [78]. In contrast, Oldford et al. found that at basal levels, C57BL/6N mice have approximately 2-fold higher hepatic mitochondrial reactive oxygen species (ROS) production compared to C57BL/6J [79]. Several differences in expression of ROS producing enzymes and shifting contributions of individual enzymes between these two substrains cause changes to the hepatic ROS pool [79]. With a high-fat diet, C57BL/6J mice have higher hepatic triglyceride burden, but less weight gain and markers of liver damage. However, both show similar levels of hepatic fibrosis [78]. These results indicate that there can be substantial differences in the pathology of NAFLD and NASH between these two substrains and emphasizes the importance of reporting specific mouse backgrounds in methods sections.

Interestingly, researchers have taken advantage of different metabolic susceptibilities across mouse strains to create more ideal models. Asgharpour et al. discovered that a stable isogenic cross between C57BL/6J and 129S1/ScImJ mice creates a strain with signs of progressive NAFLD following feeding of a diet containing 42% total kcal from fat with 23 g/L fructose and 19 g/L sucrose added to the drinking water. These DIAMOND (**D**iet-**I**nduced **A**nimal **M**odel **o**f **N**on-alcoholic fatty liver **D**isease) mice develop obesity, insulin resistance, hypertriglyceridemia, steatosis and steatohepatitis, and fibrosis by 16 weeks on the diet and spontaneously develop hepatocellular carcinoma. They also report activation of lipogenic, inflammatory and apoptotic signaling akin to human NASH. Notably, only the isogenic cross mice, and not the parental strains individually, recapitulate all the measured aspects of human NASH [80]. Given increased susceptibility to liver damage, this new genetic background is promising for NAFLD/NASH research; however, like other mouse strains, there is concern for genetic drift after multiple generations. The model is commercially available, but it appears female mice are not available from the supplier, which could complicate breeding strategies.

## 11. How Long Do I Need to Feed Mice to Promote NASH?

An important aspect of NAFLD to NASH progression is time. In humans, the transition between each stage of the disease can take between 7 and 10 years [81]. During a longitudinal study, Krishnan et al. report that insulin resistance increases 10-fold and quickly plateaus, along with hepatic lipid, after 4 weeks of a Western-style diet (17.4% protein, 50% carbohydrate, 20% fat supplemented with 23.1 g/L fructose and 18.9 g/L glucose) [56]. NASH and hepatic fibrosis are established by week 16 [56]. The authors report similar findings to Ito et al., where insulin resistance develops before hepatic inflammation and fibrosis [38,56]. Only 20% of mice on the Western-style diet develop spontaneous hepatic tumors by the end of the study (36 weeks), indicating that longer times might be needed for higher penetrance [56].

Using a similar American lifestyle-induced obesity syndrome (ALIOS) diet containing 45% fat (of which 30% is trans fat), and added liquid sugar (55% fructose, 45% glucose), Harris and colleagues monitored NASH development in male and female mice over 52 weeks. By week 26, ALIOS-fed mice have hepatic steatosis, inflammation and fibrosis [82]. By week 52, ALIOS-fed mice have significantly increased body weight, higher incidence of spontaneous hepatic tumors, increased adipose tissue mass, insulin resistance, and hyperlipidemia [82]. Male mice gain weight more slowly after 20 weeks, whereas female mice on the ALIOS diet continue to increase in size until week 52 [82]. In line with this, a study in male Wistar rats reports weight gain, hepatic steatosis, hyperinsulinemia, and hyperglycemia by 8 weeks on a high-fat, high-fructose diet but these also reach a plateau by week 12 [83].

Casagrande et al. identified that, in addition to duration of the diet, the age of the animals when the diet begins is also an important factor. The start date for a rodent diet can be arbitrary, ranging from birth (uncommon), weaning (3 weeks old) and up to 12 weeks old. When comparing Wistar rats started on a high-fat/high-fructose diet at 6 versus 12 weeks old, animals started at a younger age have higher levels of hepatic triglycerides, more weight gain, and increased IL-6 and IL-10 after 13 weeks [84]. Mice started at an older age show only increases in TNF-α. Both age groups exhibit similar increases in serum triglycerides and adiposity [84]. Collectively these results indicate that an obesogenic diet can promote NASH development after 16 weeks in mice, but for progression to NASH and cancer, it likely requires much longer diet regimes, upwards of 36 weeks.

Lastly, age of the animals also affects outcomes. Like a human, a mouse at 6 weeks old (roughly equivalent to a human teenager) has different physiology than a mouse at 12 weeks old (fully mature adult). Researchers may want to consider whether they want to model disease where diet/obesity begins in adolescence versus adulthood.

## 12. Late-Stage Endpoints

The transition from NAFLD to HCC is marked by significant inflammation, fibrosis, cirrhosis, and tumor formation. Unfortunately, promoting this transition in mice solely through diet is an ongoing challenge. A high-sugar diet is sufficient to promote hepatic fibrosis [35], with added fat [42,43] and cholesterol [50,61,85,86] increasing rate and severity of fibrosis after 20–60 weeks. While promoting late-stage NASH is possible using dietary methods (cirrhosis is rare), it requires long time periods and has unreliable transition from NASH to HCC. Zhang et al. show that feeding mice a high-fat/high-cholesterol diet (19.7% protein, 36.6% carbohydrate, 43.7% fat, 0.203% cholesterol) results in tumor development in 68% of animals after 14 months [86]. These animals have elevated levels of hepatic inflammatory cytokines and pronounced fibrosis. Interestingly, tumors are observed as early as 10 months; however, tumor incidence increases dramatically between 12 and 14 months. The DIAMOND mouse spontaneously develops HCC in approximately 90% of mice between 8 to 12 months [80]. As mentioned above, these mice also display many of the metabolic and histopathological features of advanced human NASH, including several stages of fibrosis. However, most studies in mice still model HCC using a combination of an obesogenic diet and a tumor promoting factor (e.g., genetic modification, chemotoxins, or implantations) [85]. HCC models relying solely on diet have yet to be widely used because of the extensive time required to detect tumors (>12 months) and overall low tumor rate accompanying these models.

## 13. So… What Model Should I Use to Study NAFLD/NASH in Rodents?

With the wide variety of rodent diets that one could use for NAFLD research, it can be difficult to decide what regimen is best. With recognition of a unique etiology for MAFLD, there is more emphasis on diet-induced models that reflect both liver phenotype and metabolic dysfunction. Older/historical diet formulations may not take into consideration many of the factors we now know are important.

A mix of fructose and glucose in the drinking water may be a more effective than fructose alone to promote hepatic steatosis, and sucrose can help drive hepatic inflammation. In addition to being an effective way to promote NAFLD in mice, liquid sugar allows the use of lower overall sugar concentrations and may better recapitulate how humans ingest this macronutrient.

Rodent diets with 45% kcal fat are now common and are equally effective at promoting obesity and NAFLD as higher-fat diets (e.g., 60%), while avoiding potentially confounding deficiencies in other macronutrients (i.e., protein or carbohydrate). To develop a NASH-like phenotype, cholesterol also seems important. While dietary cholesterol may no longer be a worry for cardiovascular disease, it may still damage our livers [53]. Evidence suggests that levels as low as 0.5% cholesterol in mouse chow can achieve hepatic inflammation and fibrosis seen in NASH, if it is combined with high fat and high sugar.

Control diet composition can be as important as the obesogenic diet, depending on what you are studying. Considering whether a matched control diet is more appropriate will likely allow stronger and more reproducible conclusions. At minimum, researchers should consistently report supplier and compositions of all diets used (including standard lab chow) to help interpretation and comparison of results across studies.

Lastly, two other important considerations are genetic background of the mice and duration of feeding. Although any strain can be used, there is an abundance of data supporting use of C57BL/6 as a model of NAFLD while one should pay attention to substrains, which cannot be interchanged. Essential aspects of human NAFLD are age and time. Advanced disease requires many years to develop, and this is also true in mice. Long-term feeding protocols may need to be more widely appreciated to better understand how MAFLD develops.

It should be noted that there are several differences between rodents and humans that limit our ability to recapitulate human disease in animals. Murine lipid and cholesterol metabolism differ considerably from that of humans [87,88]. Humans and mice differ also in the composition of their gut microbiome [89], which can impact digestion and metabolism of dietary components (e.g., fructose), impacting liver health [28]. These factors, and likely many other species differences in metabolism and behaviour, prevent rodent models from fully mirroring human NAFLD. However, mice remain an invaluable tool for basic research and they facilitated many discoveries that translated to humans, such as showing the benefits of GLP-1 receptor agonists and FGF21 on liver health [90,91,92].

## 14. Conclusions

Navigating the wide variety of available dietary animal models within NAFLD pre-clinical research can be overwhelming. This review is intended to present current data regarding major dietary components influencing NAFLD/NASH, with the goal of clarifying essential components and ideal quantities. Scientific questions and purposes vary widely and should ultimately direct the choice of diet. However, we encourage researchers to consider diet composition and duration as variables equal in importance to sex, age and genotype when modelling the pathology of human NAFLD/NASH.

## Figures and Tables

**Figure 1 cells-10-01805-f001:**
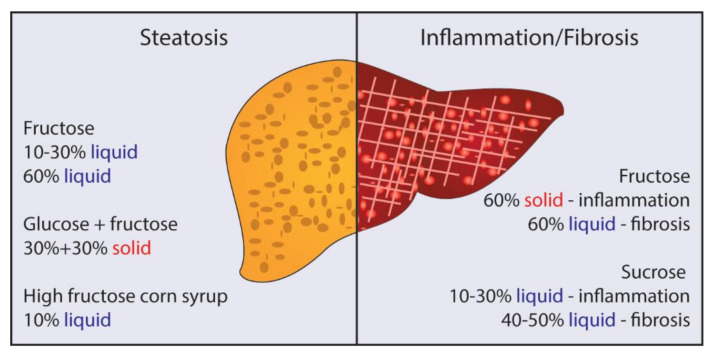
Dietary sugars promoting either hepatic steatosis or inflammation and fibrosis. Steatosis can be induced by liquid fructose (10–30% *w*/*v*) or by a mixture of glucose and fructose (30% + 30% *w/w* solid, or 10% *w*/*v* high-fructose corn syrup in water) alone or in combination with high-fat content. 60% liquid fructose induces both steatosis and fibrosis, while 60% solid fructose promotes hepatic inflammation. Liquid sucrose (10–30% *w*/*v*), with or without high fat, stimulates hepatic inflammation, with higher doses (40–50% *w*/*v*) promoting fibrosis.

**Figure 2 cells-10-01805-f002:**
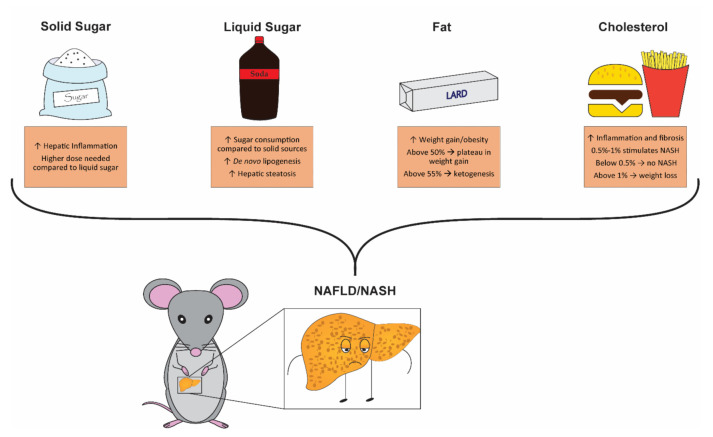
Summary of key dietary components contributions when modeling non-alcoholic fatty liver disease (NAFLD) and non-alcoholic steatohepatitis (NASH) in rodents. Inclusion of solid sugar stimulates hepatic inflammation compared to liquid sugar, whereas liquid sugar contributes more to hepatic lipid accumulation. Using solid sugars requires higher dosage to induce liver damage compared to liquid sources. High fat is required to stimulate obesity; however, at quantities above 50% kcal there is a plateau in weight gain. Inclusion of cholesterol promotes hepatic inflammation and fibrosis. Evidence suggests that the optimal dosage of cholesterol for mice is between 0.5% and 1%, but may vary between mouse strains. Below 0.5%, diets fail to induce inflammation and NASH, while above 1% prevents weight gain.

## Data Availability

Not applicable.

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
