# Peer review of "Diet-Induced Models of Non-Alcoholic Fatty Liver Disease: Food for Thought on Sugar, Fat, and Cholesterol"

_cells, 2021, doi:10.3390/cells10071805_

Round 1
Reviewer 1 Report
This review describes the influence of dietary components -sugars, fat, and cholesterol- to non-alcoholic fatty liver disease (NAFLD) in animal models. The content of the manuscript seems interesting, however it does not provide significant information of such dietary components in the progression of NAFLD at the molecular / cellular level. This reviewer advises the authors to submit this manuscript to a specialized gastroenterology / hepatology journal.
Author Response
Thank you for taking the time to read the manuscript and provide comments. We were invited by the editors to submit this piece for a special issue and the topic/content was discussed beforehand.
Reviewer 2 Report
The use of animal models for elucidation of the pathophysiology of NAFLD / NASH and the development of new drug therapies is an unavoidable issue, but there is still no animal model that can completely reproduce human NAFLD / NASH. For rodents used as animal models, attempts have been made to modify the dietary content to induce NAFLD / NASH pathology, and some of them have been successful. This review presents the trends of research to date and the ideal content that the authors consider regarding dietary components such as sugar, fat, and cholesterol given to animal models.
Comments:
The authors give detailed examples of possible dietary modification for the development for diet-induced models of NAFLD. However, as the authors state at the end of the text, this review only exemplified the content of the current various experimental diets. It takes time to develop an appropriate experimental diet, and it is difficult to determine what is the most appropriate dietary content for rodents with different nutrient metabolic pathways and intestinal bacteria from humans. The lipid metabolism including bile acid metabolism is completely different from that of humans, and that the composition of intestinal bacteria is different from that of humans except when fecal transplantation is performed on sterile mice, therefore, it should be shown that there is still a limit to creating a model based on dietary content alone even without the use of genetic modifications or toxic substances.
It is up to the reader to decide what kind of dietary content to choose, but I think this review is worth referring to what kind of dietary content should be considered when constructing a dietary NAFLD / NASH model in the future.
Author Response
We thank you for taking the time to review our work and provide comments. We also agree that mice have significant limitations to model human NAFLD and that it should be discussed. To address your concerns, we have added a short paragraph to section 13 (line 507-515, clean version) that notes some of the metabolic and physiological differences between mice and humans that prevent complete modeling of NAFLD in mice.
Round 2
Reviewer 1 Report
N/A